# Description of Transfer Processes in a Locally Nonequilibrium Medium

**DOI:** 10.3390/e21010009

**Published:** 2018-12-23

**Authors:** Andrey N. Morozov

**Affiliations:** Bauman Moscow State Technical University, 2nd Baumanskaya str., 5, Moscow 105005, Russia; amor59@mail.ru

**Keywords:** transfer processes, diffusion, irreversible processes, locally nonequilibrium medium, entropy production, flicker noise

## Abstract

This paper presents a description of the fluctuations in transfer processes in a locally nonequilibrium medium. We obtained equations which allow the fluctuations range to be determined for a transferred physical value. It was shown that the general method of describing fluctuations for the processes of diffusion, heat transfer, and viscous fluid flow can be applied. It was established that the fluctuation spectrum during the transfer processes has the character of flicker noise in the low-frequency spectral range.

## 1. Introduction

Irreversible processes, and in particular, transfer processes, are generally described in accordance with the local equilibrium principle, stating that every small element of a medium is in the equilibrium state, and the local entropy in this medium can be defined via the same thermodynamic variables function as that which is correct for the whole system in equilibrium [1]. From the physical perspective, the local equilibrium principle is valid for the cases where the chaotization period, during which the equilibrium is established in physically infinitesimal volume, is much less than the characteristic period needed for the process under discussion.

The description of transfer processes in a local non-equilibrium medium implies a certain finite delay time $\tau_{0}$ between the moment the thermodynamic force $\vec{X}$ is applied and the response in the form of a thermodynamic flow $\vec{J}$ [2,3]. The above period $\tau_{0}$ is equal to the chaotization time constant of particles of the medium [4].

The transfer processes in a medium are accompanied with entropy $\sigma_{S}$ production and the generation of flicker noise [5,6]. In particular, such noise is observed in the case of the flow of electric current in semiconductors [7,8] and in small volumes of electrolyte [9,10,11]. The main features of the flicker noise are the strong time correlation and large time memory, and these distinguish it from white noise, which occurs in equilibrium processes.

It is worth mentioning that both of the above fluctuations are of a fundamental nature and can be observed for all processes in physical media. Thereby white noise occurs in equilibrium medium without any irreversible processes, and flicker noise occurs in a locally nonequilibrium medium in the presence of irreversible processes accompanied with the production of entropy.

Flicker noise can be theoretically explained within a model assuming fluctuations of kinetic coefficients, in particular of the coefficient of viscosity with Brownian motion and diffusion, and can be mathematically described within the theory of non-Markov random processes with memory [12,13,14]. Note that in recent years the number of studies dealing with non-Markov processes has increased. They are dedicated to the following non-Markov processes: Brownian motion [15,16,17,18], abnormal diffusion [19,20,21,22,23], relaxation of paramagnetic ions in solutions [23], electronic paramagnetic resonance [24], entangled state of particles [25], quantum processes [26,27,28,29,30], turbulence [31], etc.

The aim of the present work is to develop a method for describing the transfer processes in a locally nonequilibrium medium using the model of non-equilibrium fluctuations presented in the work [4].

## 2. Transfer Processes in an Equilibrium Medium

In the following paragraphs we describe transfer processes in an equilibrium medium. Let us take Ω as a transferred physical value. Then we can present the thermodynamic force as:(1)$$\vec{X}=-\mathrm{grad}\Omega$$
and thermodynamic flow as:(2)$$\vec{J}=L_{\Omega }\vec{X}+\delta \vec{J}$$
where $L_{\Omega }$ is a kinetic coefficient and $\delta \vec{J}$ is the random thermodynamic flow resulting from equilibrium fluctuations of the medium.

It is worth mentioning that Equation (2) does not take into account the cross-coupling effects, which are of great significance for describing transfer processes in a non-equilibrium medium [32]. The proposed description can be also applied for this more general case.

For the correlation function $\langle \delta J_{i}\delta J_{k}\rangle$, the values of $\delta \vec{J}$ can be written as:(3)$$\langle \delta J_{i}\delta J_{k}\rangle =2\delta_{ik}\alpha_{\Omega }\delta \left(t_{2}-t_{1}\right)\delta \left(x_{2}-x_{1}\right)\delta \left(y_{2}-y_{1}\right)\delta \left(z_{2}-z_{1}\right)$$
where parameters i and k correspond to coordinates x, y, z, and $\delta_{ik}$ is the Kronecker delta, $\delta_{ii}=1$, ${\delta_{ik}|}_{i\neq k}=0$.

The physical value Ω depends on the process being described. When describing diffusion, the relative concentration of the additive $\delta n=n/n_{0}$ is used as the parameter Ω, the kinetic coefficient $L_{\Omega }=n_{0}Dk_{B}$, and the coefficient $\alpha_{\Omega }=L_{\Omega }k_{B}$. Here $k_{B}$ is the Boltzmann constant, $n_{0}$ is the medium’s particle concentration, and D is the diffusion coefficient.

For the case of heat conductivity, we use medium temperature T instead of Ω, the kinetic coefficient $L_{\Omega }=\lambda$, and the coefficient $\alpha_{\Omega }=L_{\Omega }T_{0}^{2}k_{B}$. Here λ is the heat conductivity coefficient and $T_{0}$ is the medium temperature without non-equilibrium fluctuations.

For descriptions of non-contractible viscous fluid at a low Reynolds numbers, we take the velocity v of molecules’ organized motion as the Ω value, the kinetic coefficient $L_{\Omega }=\eta$, and the coefficient $\alpha_{\Omega }=L_{\Omega }T_{0}k_{B}$. Here η is the viscosity coefficient.

The transfer process in an equilibrium medium can be described using the following equation:(4)$$\beta_{\Omega }\frac{\partial \Omega }{\partial t}+\mathrm{div}\vec{J}=0$$

When describing diffusion, $\beta_{\Omega }=n_{0}k_{B}$, heat conductivity $\beta_{\Omega }=c$, and viscosity flow $\beta_{\Omega }=\rho$. Here c is the heat capacity of a unit of volume of the medium; ρ is the density of the fluid.

Substitution of Equations (1) and (2) into Equation (4) gives the following:(5)$$\beta_{\Omega }\frac{\partial \Omega }{\partial t}-L_{\Omega }\left(\frac{\partial^{2}\Omega }{\partial x^{2}}+\frac{\partial^{2}\Omega }{\partial y^{2}}+\frac{\partial^{2}\Omega }{\partial z^{2}}\right)=-\frac{\partial }{\partial x}\delta J_{x}-\frac{\partial }{\partial y}\delta J_{y}-\frac{\partial }{\partial z}\delta J_{z}$$

Next, we perform the Laplace transformation for the last equation using coordinates x, y, z. We find the following:(6)$$\beta_{\Omega }\frac{\partial \tilde{\Omega }}{\partial t}-L_{\Omega }\left(u_{x}^{2}+u_{y}^{2}+u_{z}^{2}\right)\tilde{\Omega }=-u_{x}\delta {\tilde{J}}_{x}-u_{y}\delta {\tilde{J}}_{y}-u_{z}\delta {\tilde{J}}_{z}$$

Here $\tilde{\Omega }$ and $\delta {\tilde{J}}_{x,y,z}$ are representations of the corresponding functions Ω and $\delta J_{x,y,z}$, and $u_{x}$, $u_{y}$, $u_{z}$ are representations of x, y, z variables.

After the integration of the last equation, we obtain:(7)$$\tilde{\Omega }\left(t,u_{x},u_{y},u_{z}\right)=-\int_{-\infty }^{t}\exp\left[\frac{L_{\Omega }}{\beta_{\Omega }}\left(u_{x}^{2}+u_{y}^{2}+u_{z}^{2}\right)\left(t-\tau \right)\right]\left(\frac{u_{x}\delta {\tilde{J}}_{x}+u_{y}\delta {\tilde{J}}_{y}+u_{z}\delta {\tilde{J}}_{z}}{\beta_{\Omega }}\right)d\tau .$$

For the correlation function $\langle \tilde{\Omega }\left(t_{1},u_{x1},u_{y1},u_{z1}\right)\tilde{\Omega }\left(t_{2},u_{x2},u_{y2},u_{z2}\right)\rangle$ from Equation (7) after integration with Equation (3), we find the following:(8)$$\begin{array}{l} \langle \tilde{\Omega }\left(t_{1},u_{x1},u_{y1},u_{z1}\right)\tilde{\Omega }\left(t_{2},u_{x2},u_{y2},u_{z2}\right)\rangle \\ \;=-\frac{2\alpha_{\Omega }}{L_{\Omega }\beta_{\Omega }}\frac{u_{x1}u_{x2}+u_{y1}u_{y2}+u_{z1}u_{z2}}{u_{x1}^{2}+u_{y1}^{2}+u_{z1}^{2}+u_{x2}^{2}+u_{y2}^{2}+u_{z2}^{2}}exp[\frac{L_{\Omega }}{\beta_{\Omega }}(u_{x2}^{2}+u_{y2}^{2}+u_{z2}^{2})(t_{2}\\ \;-t_{1})] \end{array}$$

From Equation (8) for the dispersion of the function $\tilde{\Omega }\left(t_{1},u_{x1},u_{y1},u_{z1}\right)$, we obtain:(9)$$\langle {\tilde{\Omega }}^{2}\left(t_{1},u_{x1},u_{y1},u_{z1}\right)\rangle =-\frac{\alpha_{\Omega }}{L_{\Omega }\beta_{\Omega }}$$

For the time correlation function $\langle \tilde{\Omega }\left(t_{1}\right)\tilde{\Omega }\left(t_{2}\right)\rangle$, when $u_{x}=u_{x1}=u_{x2}$, $u_{y}=u_{y1}=u_{y2}$, $u_{z}=u_{z1}=u_{z2}$, we find:(10)$$\langle \tilde{\Omega }\left(t_{1}\right)\tilde{\Omega }\left(t_{2}\right)\rangle =-\frac{\alpha_{\Omega }}{L_{\Omega }\beta_{\Omega }}exp\left[\frac{L_{\Omega }}{\beta_{\Omega }}\left(u_{x}^{2}+u_{y}^{2}+u_{z}^{2}\right)\left(t_{2}-t_{1}\right)\right]$$

Equation (10) after the inverse Laplace transformation enables us to find the correlation function for fluctuations of the physical value Ω. For the description of diffusion, Equation (10) takes the form:(11)$$\langle \delta \tilde{n}\left(t_{1}\right)\delta \tilde{n}\left(t_{2}\right)\rangle =-n_{0}^{-1}exp\left[D\left(u_{x}^{2}+u_{y}^{2}+u_{z}^{2}\right)\left(t_{2}-t_{1}\right)\right]$$
where $\delta \tilde{n}\left(t\right)$ gives the value of δn(t).

Hence, the description of transfer processes in an equilibrium medium is reduced to the use of the conventional Langevin equations for a continuous medium [33].

## 3. Transfer Processes in a Nonequilibrium Medium

Describing transfer processes in a locally nonequilibrium medium, we use the approach given in Reference [4] for Brownian motion. For a locally nonequilibrium medium, we write Equation (2) as follows:(12)$$\vec{J}=L_{\Omega }\vec{X}+\delta \vec{J}+L_{\Omega }\int_{-\infty }^{t}\frac{1}{\sqrt{\nu_{\tau }\left(t-\tau \right)}}\frac{\partial \vec{X}\left(\tau \right)}{\partial \tau }d\tau +\int_{-\infty }^{t}\frac{1}{\sqrt{t-\tau }}\delta {\vec{J}}_{\sigma }\left(\tau \right)d\tau$$
where $\nu_{\tau }=1/\tau_{0}$ is the intensity of random fluctuations in a locally nonequilibrium medium, $\tau_{0}$ is the chaotization time constant of particles of the medium, and $\delta {\vec{J}}_{\sigma }$ is the random thermodynamic flow resulting from local non-equilibrium fluctuations of the medium.

We can write the correlation function $\langle \delta J_{\sigma i}\delta J_{\sigma k}\rangle$ of the $\delta {\vec{J}}_{\sigma }$ value as:(13)$$\langle \delta J_{\sigma i}\delta J_{\sigma k}\rangle =2\delta_{ik}\gamma_{\Omega }\delta \left(t_{2}-t_{1}\right)\delta \left(x_{2}-x_{1}\right)\delta \left(y_{2}-y_{1}\right)\delta \left(z_{2}-z_{1}\right)$$
where for the description of diffusion we take the coefficient $\gamma_{\Omega }=L_{\Omega }\sigma_{S}/n_{0}$, heat conductivity $\gamma_{\Omega }=L_{\Omega }T_{0}^{2}\sigma_{S}/n_{0}$, and viscous non-contractible fluid flow $\gamma_{\Omega }=L_{\Omega }T_{0}\sigma_{S}/n_{0}$. Here $\sigma_{S}$ is the entropy production for irreversible processes taking place in the medium, which can be found by the following equation:(14)$$\sigma_{S}=\vec{J}\cdot \vec{X}=L_{\Omega }X^{2}$$

It should be pointed out that the second and fourth components of Equation (12) are omitted in Equation (14), as they describe stochastic processes with zero mathematical expectation, and the third component is missing due to its very small values at *τ*_0_ << *t*, when compared to the first summand.

Substitution of Equation (12) into Equation (4) enables us to obtain the equation for the description of a transfer process in a locally nonequilibrium medium: (15)$$\begin{array}{l} \beta_{\Omega }\frac{\partial \Omega }{\partial t}-L_{\Omega }(\frac{\partial^{2}\Omega }{\partial x^{2}}+\frac{\partial^{2}\Omega }{\partial y^{2}}+\frac{\partial^{2}\Omega }{\partial z^{2}})\\ \;-L_{\Omega }\int_{-\infty }^{t}\frac{1}{\sqrt{\nu_{\tau }(t-1)}}\frac{\partial }{\partial \tau }(\frac{\partial_{2}\Omega (\tau )}{\partial x_{2}}+\frac{\partial_{2}\Omega (\tau )}{\partial y_{2}}+\frac{\partial_{2}\Omega (\tau )}{\partial z_{2}})d\tau \\ \;=-\frac{\partial }{\partial x}\delta J_{x}-\frac{\partial }{\partial y}\delta J_{y}-\frac{\partial }{\partial z}\delta J_{z}\\ \;-\int_{-\infty }^{t}\frac{1}{\sqrt{t-1}}(\frac{\partial }{\partial x}\delta J_{\sigma x}(\tau )+\frac{\partial }{\partial y}\delta J_{y\sigma }(\tau )+\frac{\partial }{\partial z}\delta J_{\sigma z}(\tau ))d\tau \end{array}$$

The Laplace transformation by coordinates x, y, z gives:(16)$$\begin{array}{l} \beta_{\Omega }\frac{\partial \tilde{\Omega }}{\partial t}-L_{\Omega }(u_{x}^{2}+u_{y}^{2}+u_{z}^{2})\tilde{\Omega }\\ \;-L_{\Omega }\int_{-\infty }^{t}\frac{1}{\sqrt{\nu_{\tau }(t-\tau )}}(u_{x}^{2}+u_{y}^{2}+u_{z}^{2})\frac{\partial \tilde{\Omega }(\tau )}{\partial \tau }d\tau \\ \;=-u_{x}\delta {\tilde{J}}_{x}-u_{y}\delta {\tilde{J}}_{y}-u_{z}\delta {\tilde{J}}_{z}\\ \;-\int_{-\infty }^{t}\frac{1}{\sqrt{t-\tau }}(u_{x}\delta {\tilde{J}}_{\sigma x}(\tau )+u_{y}\delta {\tilde{J}}_{\sigma y}(\tau )+u_{z}\delta {\tilde{J}}_{\sigma z}(\tau ))d\tau \end{array}$$

Then, if we conduct the Laplace transformation on Equation (16) by the time variable t, we obtain:(17)$$\begin{array}{l} \left(\beta_{\Omega }s+A\left(1+\sqrt{\frac{\pi s}{\nu_{\tau }}}\right)\right)\hat{\tilde{\Omega }}\left(s\right)\\ \;=-\left(u_{x}\delta {\hat{\tilde{J}}}_{x}+u_{y}\delta {\hat{\tilde{J}}}_{y}+u_{z}\delta {\hat{\tilde{J}}}_{z}\right)-\sqrt{\frac{\pi }{s}}\left(u_{x}\delta {\hat{\tilde{J}}}_{\sigma x}+u_{y}\delta {\hat{\tilde{J}}}_{\sigma y}+u_{z}\delta {\hat{\tilde{J}}}_{\sigma z}\right) \end{array}$$
where s is a representation of the time variable t, and the A parameter is equal to:(18)$$A=-L_{\Omega }\left(u_{x}^{2}+u_{y}^{2}+u_{z}^{2}\right)$$

Equation (17) can be represented as:(19)$$\hat{\tilde{\Omega }}\left(s\right)=-\frac{\left(u_{x}\delta {\hat{\tilde{J}}}_{x}+u_{y}\delta {\hat{\tilde{J}}}_{y}+u_{z}\delta {\hat{\tilde{J}}}_{z}\right)+\sqrt{\frac{\pi }{s}}\left(u_{x}\delta {\hat{\tilde{J}}}_{\sigma x}+u_{y}\delta {\hat{\tilde{J}}}_{\sigma y}+u_{z}\delta {\hat{\tilde{J}}}_{\sigma z}\right)}{\left(\beta_{\Omega }s+A\left(1+\sqrt{\frac{\pi s}{\nu_{\tau }}}\right)\right)}$$

The inverse Laplace transformation by $u_{x}$, $u_{y}$, $u_{z}$, and s allows us to obtain the equation for fluctuations of the transported physical value Ω(t,x,y,z).

Equation (19), obtained using:(20)$$G_{\tilde{\Omega }}\left(\omega \right)={\left|\hat{\tilde{\Omega }}\left(i\omega \right)\right|}^{2}$$
enables us to write the equation for the two-sided spectral density of the fluctuation of the transferred physical value $G_{\tilde{\Omega }}\left(\omega \right)$:(21)$$G_{\tilde{\Omega }}\left(\omega \right)=-\frac{2A\left(\alpha_{\Omega }+\frac{\pi \gamma_{\Omega }}{\omega }\right)}{\beta_{\Omega }^{2}\omega^{2}+A^{2}\left(1+\frac{\pi \omega }{\nu_{\tau }}\right)+A\sqrt{\frac{2\pi \omega }{\nu_{\tau }}}\left(\beta_{\Omega }\omega +A\right)}$$

When obtaining Equation (21), we took into account Equations (3) and (13) for correlation functions of the fluctuation of flows.

If we introduce the following designations:(22)$$B=\frac{2A\alpha_{\Omega }}{\beta_{\Omega }^{2}}$$
(23)$$f_{L}=\frac{A}{\beta_{\Omega }}$$
(24)$$f_{\sigma }=\frac{\gamma_{\Omega }}{\alpha_{\Omega }}$$
then the spectral density $G_{\tilde{\Omega }}\left(\omega \right)$ can be represented as:(25)$$G_{\tilde{\Omega }}\left(\omega \right)=-\frac{B\left(1+\frac{\pi f_{\sigma }}{\omega }\right)}{\omega^{2}+f_{L}^{2}\left(1+\frac{\pi \omega }{\nu_{\tau }}\right)+f_{L}\sqrt{\frac{2\pi \omega }{\nu_{\tau }}}\left(\omega +f_{L}\right)}$$

Figure 1 shows diagrams of the spectral density $G_{\tilde{\Omega }}\left(\omega \right)$ for different levels of non-equilibrium of the medium, which is characterized by the value $f_{\sigma }$.

Clearly, in the high frequency areas they coincide with each other, and in the area of low frequency an obvious inversely proportional increase of fluctuations with a simultaneous decrease of the frequency ω can be observed. Moreover, the intensity of these fluctuations is linearly dependent on the value $f_{\sigma }$, and consequently on the production of entropy $\sigma_{S}$.

For low frequencies, when $\omega <<A/\beta_{\Omega }$ and $\omega <<\nu_{\tau }$, Equation (21) takes the form:(26)$$G_{\tilde{\Omega }}\left(\omega \right)=-\frac{2\alpha_{\Omega }}{L_{\Omega }\left(u_{x}^{2}+u_{y}^{2}+u_{z}^{2}\right)}\left(1+\frac{\pi \sigma_{S}}{n_{0}k_{B}\omega }\right)$$
taking into account that:(27)$$\frac{\gamma_{\Omega }}{\alpha_{\Omega }}=\frac{\sigma_{S}}{n_{0}k_{B}}$$

The last item within the brackets of Equation (26) becomes more significant for the following frequency:(28)$$\omega >\frac{\pi \sigma_{S}}{n_{0}k_{B}}=\frac{\pi L_{\Omega }X^{2}}{n_{0}k_{B}}$$

For the case of diffusion, Equation (28) takes the form:(29)$$\omega >\pi DX^{2}$$

If diffusion takes place in the air with characteristic values of the diffusion coefficient $D=10^{-5}$ m^2^s^−1^ and X=1 m^−1^, then we have to measure the spectral density of fluctuations of diffusing elements concentration at very low frequencies, which are close to $\omega \approx 10^{-5}$ s^−1^, to clearly observe flicker noise. 

## 4. Conclusions

The method presented for the description of transfer processes in a locally nonequilibrium medium allows us to determine the spectral density of fluctuations of a transferred physical value, as well as to establish that its spectrum in the low-frequency range has a flicker-noise nature. The method described can be used not only for the study of diffusion, heat conductivity, and viscosity flow phenomena, but also for any other transfer process.

Unlike Reference [4], which analyses a case for the Brownian motion described using an integro-differential equation with a substantial derivative, this work presents descriptions of transfer processes in a continuum non-equilibrium medium, which is described with integro-differential equations with partial derivatives. In this regard, the results obtained reflect the specific character of the transfer processes described.

## Figures and Tables

**Figure 1 entropy-21-00009-f001:**
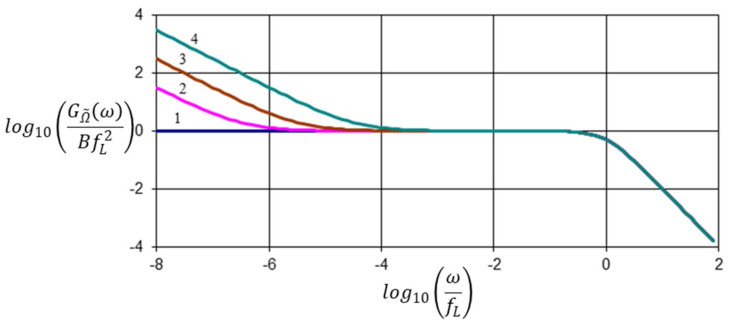
Spectral density $G_{\tilde{\Omega }}\left(\omega \right)$ for different values of $f_{\sigma }$: 1—0$f_{L}$; 2—10^−7^$f_{L}$; 3—10^−6^$f_{L}$; 4—10^−5^$f_{L}$.

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
