# Peer review of "Description of Transfer Processes in a Locally Nonequilibrium Medium"

_entropy, 2018, doi:10.3390/e21010009_

Reviewer 1 Report

Report on the manuscript “Description of Transfer Processes in a Locally Nonequilibrium Medium” by A.Morozov

Transport processes in non-equilibrium media are considered on the basis of the approach proposed recently by the same author. The method allows determining the spectral density of fluctuations. It is shown that under non-equilibrium conditions, the fluctuation spectrum corresponds to the flicker noise. The method is applied to study diffusion, heat conduction and viscosity, but it can be further extended for other transport processes.

The paper is well written, contains new original results, its subject is appropriate for the Entropy journal. It can be accepted after the author addresses the comments listed below.

Comments:

1. In Eq. (2), the cross-coupling effects are neglected, which can be of importance in non-equilibrium media, see for instance [E.Kustova and D.Giordano, Cross-coupling effects in chemically non-equilibrium viscous compressible flows, Chem. Phys. 379(1–3), 83–91 (2011)]. A comment on the possible impact of cross-coupling effects on the applicability of the proposed method would be rather useful.

2. Equations (1) and (2) are written for vector forces and fluxes. However, the forces and fluxes can be of any tensorial order: vectors, tensors or scalars (for instance, chemical reaction rates). To stay general, it is better to use a unique notation, allowing for any forces/fluxes.

3. In fluid dynamics, the notation ?/?? is basically used for the partial time derivative, and not for the substantial derivative, which is ???=???+?∙∇. In Eq. (4), does the author mean the time derivative or the substantial derivative? Please explain.

4. In Eq. (14), if vector forces/fluxes are meant, the scalar product sign has to be added. Moreover, it is not clear why the 2-d, 3-d and 4th terms from Eq. (12) vanish in Eq. (14). Please provide some explanation.

5. In 124, it is written ‘The last item within the brackets of the equation (27)…’ However, Eq. (27) does not include any brackets.

6. In lines 126-129, the dimensions are given in Russian. Please translate into English.

Author Response

Dear Reviewer, 

The author sincerely appreciates your kind consideration of the presented work and your comments.

In response to the remarks made, I would like to clarify the following. 

1.     The work [E. Kustova and D. Giordano, Cross-coupling effects in chemically non-equilibrium viscous compressible flows, Chem. Phys. 379(1–3), 83–91 (2011)] was added to the list of references. The cross-coupling effects can be taken into account as part of this method, which results in relative summands in equation (12). A correspondent remark has been added into the text (lines 54–56).

2.     The article discusses only vector fluxes and forces. This is aimed at simplification of mathematical manipulations. Indeed, the proposed approach can be applied to the general case, which includes description of tensorial fluxes and forces, as well as scalar ones. But this requires considering specifics of the mathematical description of the mentioned fluxes and forces.

3.     In the equation (4), the author uses a partial derivative, not a substantial one, as fluid dynamics description is believed to be performed at low Reynolds numbers, and flux components are not to be taken into account. This is mentioned in the article (line 67).

4.     The scalar product sign has been added into the equation (14). The second and fourth components of the equation (12) are missing in equation (14), as they describe  stochastic processes with zero mathematical expectation, and the third component is missed due to its very low values at τ0<<t, when compared to the first summand. The text of the article was complemented with corresponding remarks (lines 102–105).

5.     In line 124 (131 in the new version of the article), the reference (27) was replaced with (26). It was an error.

6.     English dimensions were added to the text (lines 134-137).

I hope I have managed to provide a satisfactory answer to your remarks. 

Sincerely,

Andrey N. Morozov

Reviewer 2 Report

This article presents a calculation of the spectral density of a quantity in a nonequilibrium system. Fluctuations are introduced through the currents in a way similar to that of fluctuating hydrodynamics. The novelty is that the medium is dispersive: transport coefficients depend on the frequenc. The analysis in then similar to that of a Brownian particle with a frequency dependent friction coefficient. It is shown that the spectral density shows flicker noise in a certain frequency range. 

The article is based on a previuosly published article (Ref. 4 of the manuscript). Authors should show how different are the results obtained from those of the reference mentioned.

Author Response

Dear Reviewer, 

The author sincerely appreciates your kind consideration of the presented work and your comments.

In response to the remarks made, I would like to clarify the following.

The article [4] analyses a case for the Brownian motion described with the help of an integro-differential equation with an substantial derivative. The proposed work presents descriptions of transfer processes in a continuum non-equilibrium medium, which is described with integro-differential equations with partial derivatives. Furthermore, the article is intended to develop the description  which can be applied to various transfer processes. As this takes place, the results obtained reflect the specific character of the transfer processes described. A comment has been added to the body of the text (lines 143–148).

I hope I have managed to provide a satisfactory answer to your remarks. 

Sincerely,

Andrey N. Morozov

Round  2

Reviewer 2 Report

The author has explained my question satisfactorily. The article is now suitable for publication.